# DATA SELECTION AND ACTIVE LEARNING VIA LOW-RANK APPROXIMATION

## ABSTRACT

In the data selection problem, the objective is to choose a small, representative subset of data that can be used to efficiently train a machine learning model. Sener and Savarese [ICLR 2018] showed that given an embedding representation of the data and certain underlying geometric assumptions, $k$-center clustering heuristics can be employed to perform data selection. This notion was further explored by Axiotis et. al. [ICML 2024], who proposed a data selection approach based on $k$-means clustering and sensitivity sampling. However, these approaches all assume the datasets intrinsically exhibit certain geometric properties that can be captured by clustering, whereas a large number of datasets actually possess algebraic structure that are better utilized by low-rank approximation, feature selection, or principal component analysis. In this paper, we introduce a new data selection technique based on low-rank approximation and residual sampling. Given an embedding representation of the data with specific assumptions, which intuitively correspond to algebraic or angular notions of Lipschitzness, we give a method that selects roughly $k + \frac{1}{\varepsilon^2}$ items whose average loss approximates the average loss of the entire dataset, up to a relative $(1 + \varepsilon)$ error and an additive $\varepsilon\Phi_k$ term, where $\Phi_k$ denotes the optimal rank-$k$ cost for fitting the input embedding. We complement our theoretical guarantees with empirical evaluations, showing that for a number of important real-world datasets, our data selection approach outperforms previous strategies based on uniform sampling or sensitivity sampling.

## 1 INTRODUCTION

The unprecedented growth of both datasets and models has fueled the success of modern machine learning, culminating in foundation models with remarkable capabilities across domains. Yet, this success comes at a steep cost: training and fine-tuning these models requires immense computational resources, extensive datasets, and long training cycles, rendering the process nearly impossible for most academic groups and small-scale companies. Importantly, however, it is now well understood that using the entire dataset is rarely necessary—carefully chosen subsets of data often suffice to achieve nearly the same performance, with only a marginal increase in error. This observation raises a fundamental and urgent question: *how can we efficiently identify the most informative subset of training data without compromising model quality?*

While uniform sampling can often perform reasonably well in practice, it is inherently suboptimal on complex, imbalanced, or redundant datasets. To better capture the utility of individual data points for training, a large body of work on *data selection and active learning* aims to identify examples that are most informative given their uniqueness, quality, or relationship to the model's current knowledge. Although no universally optimal active learning strategy exists (Dasgupta, 2004), many heuristics have proven effective in practice (Settles, 2009; Ren et al., 2020). Active learning is typically framed as an iterative process: a model is trained, then used to score and select a subset of unlabeled points for annotation. State-of-the-art methods rely on *uncertainty-based criteria*, such as margin or entropy, which prioritize points on which the model is least confident. However, when applied at scale, such strategies face two key limitations. First, computing selection scores requires evaluating the model on the entire dataset, which is computationally prohibitive for modern large-scale architectures. Second, practical training pipelines, especially those involving CNNs or foundation models, require acquiring and processing data in *batches* rather than one point at a time.

This induces correlations among selected samples, which substantially reduces the effectiveness of uncertainty-based heuristics and limits their impact on training efficiency.

A major step forward was made by Sener & Savarese (2018), who reframed active learning as a *coreset selection* problem. Their insight was that the difficulty of applying uncertainty-based methods in modern training pipelines stems from two central obstacles:

(1) First, training must proceed in *batches*, not one example at a time. Effective batch acquisition requires both informativeness and diversity, yet diversity often conflicts with standard objectives such as margin maximization, leading to redundant or near-duplicate selections.

(2) Second, computing uncertainty scores requires running inference over the entire dataset, which is already prohibitive for CNNs and becomes intractable for current large-scale architectures.

To overcome these barriers, Sener & Savarese (2018) proposed to directly seek a small subset of data that serves as a coreset: training on the coreset should yield nearly the same model as training on the full dataset. In formal terms, the gradients (or losses) averaged over the coreset should approximate those of the entire dataset, so that optimization on the subset faithfully reproduces the effect of optimization on all data. Since computing these gradients exactly is impractical, they introduced a geometric relaxation: given an embedding representation of the data, one can approximate the coreset by solving a variant of the classical $k$-center problem. This formulation is both natural and widely applicable, as embeddings can be obtained from pretrained encoders such as BERT (Devlin et al., 2019a), word2vec (Mikolov et al., 2013), GloVe (Pennington et al., 2014), ResNet (He et al., 2016), or CLIP (Radford et al., 2021). Empirically, this approach delivered state-of-the-art results in image classification benchmarks, demonstrating that geometric coreset selection can substantially outperform uncertainty-based heuristics in batch training scenarios.

Unfortunately, while embedding-based selection methods based on $k$-center and its $k$-clustering variants can be effective in some settings, they often exhibit critical limitations in modern machine learning applications. In high-dimensional datasets, clustering focuses on local groupings of points and can fail to capture the dominant directions of variance, meaning that selected subsets may miss the most informative components of the data. Low-rank approximation methods, by contrast, explicitly aim to preserve these dominant directions, ensuring that small subsets retain the essential spectral structure of the dataset. This phenomenon has been recently observed in the context of Low-Rank Adaptation (LoRA) (Hu et al., 2022; Xu et al., 2024; Wu et al., 2024; Li et al., 2024), where low-rank updates capture the most important components of the parameter space and enable efficient adaptation, whereas naive clustering of embeddings may overlook key directions. These observations suggest that, while clustering retains value in some cases, low-rank-based selection can provide a more reliable foundation for data-efficient training on large-scale models. Indeed, for many foundational tasks in data analysis and machine learning, the central loss function can be expressed as a *low-rank approximation* objective. Examples include principal component analysis (PCA), matrix completion, and dimensionality reduction.

## 2 METHODOLOGY AND CONTRIBUTIONS

Fine-tuning a Large Language Model (LLM) for a specialized task, such as translation, can be extremely costly when using the entire dataset, even if ample data is available. In practice, it is often preferable to select a small subset of points that preserves the essential structure of the data while still allowing the model to achieve high performance. Directly computing data importance, for example through model-based loss or margin scores, is typically impractical because it requires evaluating every data point with the full LLM, which is computationally expensive.

In this work, we propose a framework for *data selection under low-rank losses* that addresses this challenge by combining accurate but costly scores on a small fraction of the data with fast-to-compute embeddings or sketches that capture the dominant directions of the dataset. Surprisingly, simple embeddings derived from pre-trained models, such as BERT (Devlin et al., 2019a) or word2vec (Mikolov et al., 2013), are often sufficient to approximate the low-rank structure relevant for selecting informative points, even for much larger target models. We construct a low-rank sketch of the dataset to estimate leverage scores, which informally quantify the importance of each point with respect to the orthogonal space of the sketch, and then sample rows proportionally to these

scores. This ensures that the selected subset reflects the main directions of variance, rather than merely promoting diversity as in clustering-based coresets.

By focusing on preserving the dominant spectral components, this framework offers a robust and efficient alternative to both naive subsampling and clustering-based data selection in high-dimensional and large-scale settings. Beyond the theoretical guarantees, the approach is simple, scalable, and broadly applicable. We demonstrate its effectiveness empirically on both a standard tabular dataset and challenging Llama3-8B (Dubey et al., 2024) fine-tuning on three tasks, outperforming the accuracy of existing baselines.

To derive theoretical guarantees for our low-rank sampling framework, we assume the following smoothness condition on the loss function $\ell$ and the low-dimensional factor $V$. Let $V = \mathrm{span}\{v_1, \ldots, v_k\}$ be a $k$-dimensional subspace, for instance corresponding to the top singular vectors, principal components, or basis directions from a low-rank factorization. For any point $y \in \mathbb{R}^d$, decompose

$$y = \alpha_1 v_1 + \cdots + \alpha_k v_k + r(y), \quad r(y) = \mathrm{Proj}_{V^\perp}(y),$$

where $\alpha_i = \langle y, v_i \rangle$ and $r(y)$ is the component orthogonal to $V$, i.e., the projection of $y$ onto $V^\perp$. Let $v(y) = \mathrm{Proj}_V(y) = \alpha_1 v_1 + \cdots + \alpha_k v_k$.

**Assumption 2.1.** *We assume there exist constants $\lambda, \gamma > 0$ such that*

$$|\ell(y) - \ell(v(y))| \leq \lambda \|r(y)\|_2^2, \qquad |\ell(v(y)) - (\alpha_1^2 \ell(v_1) + \cdots + \alpha_k^2 \ell(v_k))| \leq \gamma \sum_{i=1}^k |\alpha_i^2 - 1| \, \ell(v_i).$$

Intuitively, this condition decomposes the loss at $y$ into two components: a weighted sum of the losses along each basis direction $v_i$, with weights $\alpha_i^2$, and a penalty proportional to the squared norm of the component orthogonal to $V$, $\|r(y)\|_2^2$.

These assumptions are natural in many machine learning settings. For example, in low-rank regression, PCA, or matrix completion, the dominant directions of the data capture most of the variance, while deviations along the orthogonal directions contribute minimally to the loss. In LLM fine-tuning or embedding-based models, top singular vectors often align with the most informative components, and residual directions carry less signal. Similar behavior is observed in low-rank adaptation techniques such as LoRA (Hu et al., 2022; Xu et al., 2024; Wu et al., 2024; Li et al., 2024), where trainable low-rank matrices capture the key directions in the parameter space. This indicates that many real-world datasets are approximately low-rank, making these assumptions a reasonable abstraction for constructing coresets and selecting informative data efficiently. Then our main theorem is as follows:

**Theorem 2.2.** *[Coreset Guarantee for Loss Approximation] Let $D$ be a dataset of $n$ points with an embedding $E$, and suppose the loss function $\ell$ satisfies Assumption 2.1 with constants $\gamma, \lambda$. Let*

$$\Phi_k(D) = \min_{\substack{D_k \in \mathbb{R}^{n \times m} \\ \mathrm{rank}(D_k) \leq k}} \|D - D_k\|_F^2$$

*denote the best rank-$k$ approximation cost of $D$. Then there exists a randomized algorithm that constructs a weighted subset $S \subseteq D$ of size $s = \mathcal{O}\left(\frac{1}{\varepsilon^2}\right)$ with weights $w(x)$ such that, with probability at least $0.9$,*

$$\left| \sum_{x \in D} \ell(x) - \sum_{x \in S} w(x) \, \ell(x) \right| \leq \varepsilon \left( \sum_{x \in D} \ell(x) + \gamma \|D\|_F^2 + \gamma k |D| \max \ell + 2\lambda \, \Phi_k(D) \right). \quad (1)$$

*Equivalently, the weighted average loss on $S$ is within a $(1 \pm \varepsilon)$ factor of the true average loss, up to an additive term proportional to $\Phi_k(D)/n$.*

Theorem 2.2 formalizes the intuition that a small, carefully selected subset of data can effectively represent the loss of the entire dataset under low-rank structure assumptions. The theorem guarantees that a weighted subset $S$ of size $\mathcal{O}\left(\frac{1}{\varepsilon^2}\right)$ suffices to approximate the total loss over $D$ within a factor of $(1 \pm \varepsilon)$, up to an additive term proportional to $\Phi_k(D)$, the optimal rank-$k$ approximation error. The additive error in the theorem depends on $\Phi_k(D)$, the optimal low-rank approximation error. Datasets that are nearly low-rank yield smaller $\Phi_k(D)$, and hence the coreset more accurately

preserves the total loss. This also indicates a tradeoff analogous to clustering: if the data contains significant outliers or high-rank noise, the bound increases, reflecting the inherent difficulty of representing such datasets with few points. Unlike clustering-based methods, however, the low-rank approach explicitly targets directions of high variance and information content, making it more robust in high-dimensional or unbalanced settings. Practically, this result implies that training or fine-tuning models on $S$ incurs minimal loss in accuracy while substantially reducing computational cost. Our experiments show that using subsets constructed via Theorem 2.2 achieves competitive or superior performance compared to existing uniform or sensitivity sampling-based selection methods, providing a new practical sampling strategy for active regression (Chen & Price, 2019; Chen & Derezinski, 2021; Parulekar et al., 2021; Musco et al., 2022; Woodruff & Yasuda, 2023).

## 3 PROBLEM DEFINITION

### 3.1 BATCH DATA SELECTION AND LOSS DECOMPOSITION

We formally define the batch data selection problem in the context of low-rank losses. Let $D = \{(x_i, y_i)\}_{i=1}^n$ be a dataset sampled i.i.d. from a distribution $\mathcal{P}$ over $\mathcal{X} \times \mathcal{Y}$. Given a sample $x$ and its label $y$, an algorithm $\mathcal{A}$ trains a model to produce a predicted label $\hat{y}$, and incurs a loss based on the discrepancy between $y$ and $\hat{y}$. We denote this loss by $\ell(x, y; \mathcal{A})$. The goal is to select a subset $S \subseteq D$ of size at most $s$ and associate a weight function $w : S \to \mathbb{R}^+$ such that

$$\Delta(S) := \left| \sum_{i=1}^n \ell(x_i, y_i; \mathcal{A}) - \sum_{x \in S} w(x)\, \ell(x, y; \mathcal{A}) \right|$$

is minimized, while keeping the number of expensive model evaluations (i.e., queries to $\ell$) small. Observe that the expected loss of $\mathcal{A}$ can be decomposed as

$$\mathbb{E}_{(x,y)\sim\mathcal{P}}\ell(x, y; \mathcal{A}) \leq \underbrace{\left| \mathbb{E}_{(x,y)\sim\mathcal{P}}\ell(x, y; \mathcal{A}) - \frac{1}{n}\sum_{i=1}^n \ell(x_i, y_i; \mathcal{A}) \right|}_{\text{Generalization Error}} + \underbrace{\frac{1}{|C|}\sum_{j\in C}\ell(\widetilde{x}_j, \widetilde{y}_j; \mathcal{A})}_{\text{Training Error}}$$

$$+ \underbrace{\left| \frac{1}{n}\sum_{i=1}^n \ell(x_i, y_i; \mathcal{A}) - \frac{1}{|C|}\sum_{j\in C}\ell(x_j, y_j; \mathcal{A}) \right|}_{\text{Coreset Loss}},$$

where $\widetilde{S} = \{(\widetilde{x}_j, \widetilde{y}_j)\}_{j\in C}$ is a coreset constructed from $S$ via some algorithm $\mathcal{A}$. This decomposition clarifies the sources of error in batch selection: the generalization error measures the gap between empirical and population loss, the training error captures how well the model fits the selected coreset, and the coreset loss quantifies how faithfully the coreset approximates the full dataset. Our low-rank sampling strategy explicitly targets minimizing the coreset loss while requiring only a small number of expensive evaluations of $\ell$, ensuring efficient and effective model training.

We remark that like Axiotis et al. (2024), our formulation of data selection allows for *weighted sampling*, where each selected point can carry an individual weight $w(x)$, rather than assuming uniform weights like Sener & Savarese (2018). This is natural in the low-rank setting, where sampling probabilities derived from leverage scores or spectral sensitivities inherently produce non-uniform contributions to the coreset. Furthermore, rather than focusing on the loss after retraining the model on the subset, we consider the current model loss $\ell(x, y; \mathcal{A})$. Intuitively, if a subset $S$ approximates the loss of the full dataset well under the current model, it contains representative points that capture the dominant directions of the data. This allows subsequent training on $S$ to closely approximate training on the entire dataset without requiring assumptions on the label distribution or zero training loss. By framing the problem in this way, we can provide strong theoretical guarantees for low-rank coresets while maintaining flexibility and applicability to a wide range of models and loss functions.

### 3.2 ADDITIONAL PRELIMINARIES

Let $V = \text{span}\{v_1, \ldots, v_k\} \subset \mathbb{R}^d$ be a $k$-dimensional subspace with an orthonormal basis $\{v_1, \ldots, v_k\}$. For the standard inner product $\langle x, v_i \rangle$, the projection of a vector $x \in \mathbb{R}^d$ onto $V$

is defined as

$$\text{Proj}(x, V) := \sum_{i=1}^{k} \langle x, v_i \rangle v_i.$$

Let $A \in \mathbb{R}^{n \times d}$ be a data matrix. The *singular value decomposition (SVD)* of $A$ is $A = U\Sigma V^\top$, where $U \in \mathbb{R}^{n \times n}$ and $V \in \mathbb{R}^{d \times d}$ are orthogonal matrices containing the left and right singular vectors, respectively, and $\Sigma \in \mathbb{R}^{n \times d}$ is a diagonal matrix with non-negative singular values $\sigma_1 \geq \sigma_2 \geq \ldots \geq 0$.

For a target rank $k < \min(n, d)$, the *best rank-$k$ approximation* of $A$ in the Frobenius norm is obtained by truncating the SVD to the top $k$ singular values $A_k = U_k \Sigma_k V_k^\top$, where $U_k \in \mathbb{R}^{n \times k}$, $V_k \in \mathbb{R}^{d \times k}$, and $\Sigma_k \in \mathbb{R}^{k \times k}$ contain the top $k$ singular vectors and singular values. The Eckart-Young-Mirsky theorem (Eckart & Young, 1936) guarantees that

$$A_k = \arg \min_{\substack{B \in \mathbb{R}^{n \times d} \\ \text{rank}(B) \leq k}} \|A - B\|_F^2,$$

i.e., $A_k$ is the unique rank-$k$ matrix that minimizes the squared Frobenius norm of the approximation error. The optimal cost is often denoted by

$$\Phi_k(A) := \min_{\substack{B \in \mathbb{R}^{n \times d} \\ \text{rank}(B) \leq k}} \|A - B\|_F^2 = \sum_{i=k+1}^{\min(n,d)} \sigma_i^2.$$

This low-rank approximation captures the most significant directions of variance in the data, and forms the foundation for coresets and data selection under low-rank losses.

### 3.3 Algorithm: Sensitivity Sampling for Low-Rank Loss Approximation

To efficiently construct the low-rank approximation coreset, we propose an algorithm that leverages sensitivity sampling based on the low-rank structure of $D$. Instead of clustering, we use low-rank approximation techniques (e.g., via singular value decomposition) to compute importance scores for data points. From the above assumptions, we show that a carefully selected small subset of data, constructed via sensitivity sampling based on low-rank structure, provides a provably accurate approximation of the overall loss. The proof of Theorem 2.2 leverages the low-rank structure of the dataset to construct an importance-weighted coreset. The key idea is to decompose each data point $x$ into two components: its projection $v(x)$ onto a low-rank subspace $V$, and its residual $r(x)$ orthogonal to $V$. The Lipschitz-like and basis decomposition assumptions ensure that the loss $\ell(x)$ can be tightly approximated by the contributions along the basis directions plus a small penalty for the residual. Intuitively, this means that the dominant directions of variance capture most of the loss, while the orthogonal directions contribute only a limited, controllable amount.

Using this decomposition, the algorithm defines a sensitivity score for each point, reflecting how much it contributes to the total loss relative to its projection and residual. Sampling points proportionally to these scores ensures that high-impact points are more likely to be included in the coreset. By weighting the sampled points appropriately, the resulting estimator becomes unbiased. A standard concentration inequality is then used to bound the deviation of the weighted sum from the total loss, giving the high-probability guarantee. Overall, the proof formalizes the intuition that a small, carefully weighted subset of points suffices to approximate the loss of the entire dataset, with an additive term proportional to the optimal rank-$k$ approximation error $\Phi_k(D)$. We defer the full proof of Theorem 2.2 to Appendix B.

## 4 Real-World Dataset Experiments

### 4.1 Credit card dataset

We evaluate on the Default of Credit Card Clients dataset (Yeh, 2016; Yeh & Lien, 2009), which contains 30 000 records described by 23 attributes, including six months of previous bill statements, repayment statuses, credit limits and demographic variables. The binary label represents default on

---

**Algorithm 1** Sensitivity Sampling for Low-Rank Loss Approximation

---

**Input:** Dataset $D = \{x_1, \ldots, x_n\}$; target rank $k$; error parameter $\varepsilon > 0$; Constants $\lambda, \gamma$ corresponding to Assumption 2.1.

**Output:** A weighted subset $S \subseteq D$ of size $s$ that approximates the total loss.

1: Compute a rank-$k$ approximation $V$ of $D$ (e.g., via SVD) and let $v_1, \ldots, v_k$ be a basis for $V$
2: For each point $x \in D$, compute the residual vector

$$r(x) \leftarrow x - \text{Proj}(x, V).$$

3: Let $\text{Proj}(x, V) = \alpha_1 v_1 + \ldots + \alpha_k v_k$
4: $\sigma(x) \leftarrow (\gamma + 1)(\alpha_1^2 \ell(v_1) + \ldots + \alpha_k^2 \ell(v_k)) + \gamma k \xi + \lambda \|r(x)\|_2^2$,
5: Normalize the scores to obtain a probability distribution:

$$p(x) = \frac{\sigma(x)}{\sum_{y \in D} \sigma(y)}.$$

6: Set $s \leftarrow \left\lceil \varepsilon^{-2}\left(2 + \frac{2\varepsilon}{3}\right) \right\rceil$.
7: Sample $s$ points from $D$ independently according to $\{p(x)\}_{x \in D}$.
8: **for** each sampled point $x$ **do**
9:     Set its weight $w(x) \leftarrow \frac{1}{s\,p(x)}$.
10: **return** $S$ with associated weight function $w(\cdot)$.

---

the next month's payment (22% positive rate). This heterogeneous, imbalanced dataset is a standard benchmark for subsampling and downstream classification.

**Experimental setup.** All coreset experiments begin by loading the full dataset, renaming the column `default payment next month` to `Class`, dropping the `ID`, and applying z-score normalization to all 23 feature columns. We then compute the per-point squared norms $\ell_i = \|x_i\|_2^2$ and the true sum $L_{\text{true}} = \sum_{i=1}^{n} \ell_i$. We vary coreset size $s \in \{1000, 2000, 3000, 4000, 5000\}$ and repeat 100 independent trials of each method to average results.

For random sampling we draw $s$ points *with replacement* uniformly at random and assign each weight $n/s$. For clustering, we run a K-Means++ algorithm with maximum 300 iterations. We repeat the clustering 10 times and pick the best results. on the standardized data, select the nearest training point to each centroid, and weight it by its cluster size. For sensitivity sampling (Algorithm 1), we run `TruncatedSVD(n_components=5)` from scikit-learn (Pedregosa et al., 2011) to obtain projection vectors, compute projected-loss term $(\gamma + 1)\,\alpha^2 \ell(v_i)$, basis-loss term $\gamma, k, \xi$, and residual-loss term $\lambda \|r_i\|^2$ with parameters $\gamma = 5$, $\lambda = 1$, smoothing $10^{-6}$, normalize to probabilities $p_i$, sample $s$ points *with replacement* according to $p_i$, and assign weights $1/(s\,p_i)$.

In the coreset-error experiment (Figure 1a) we measure error $= \left| \sum_j w_j \|x_j\|_2^2 - L_{\text{true}} \right|$, for each trial and average across trials. In the downstream accuracy experiment (Figure 1b) we first fit a full logistic regression model (`solver='liblinear'`, `class_weight='balanced'`, `max_iter=1000`) on the training set to obtain per-point logistic losses for sensitivity sampling, then for each $s$ and each method train a logistic model with identical hyperparameters on the weighted coreset and evaluate test accuracy on the held-out 20%.

**Results and discussion.** Figure 1a shows that sensitivity sampling results in the lowest approximation error at every sample size, reducing error by an order of magnitude relative to random sampling and by roughly 50% compared to clustering at $s = 1000$, with all methods converging as $s$ increases. Figure 1b then shows that logistic regression trained on sensitivity coresets attains up to 74% test accuracy at $s = 5000$, clustering coresets reach around 70%, and random sampling only about 67%. These results confirm that the low-rank sensitivity algorithm not only tightens coreset-error bounds but also translates into improved predictive performance on an imbalanced, real-world financial dataset.

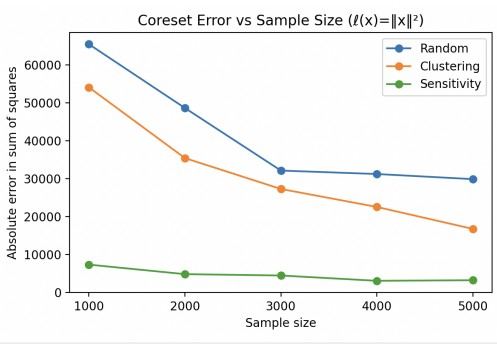
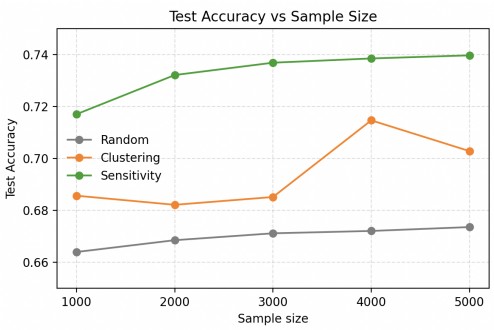

(a) Absolute error in sum of squares vs. sample size.       (b) Test accuracy vs. sample size.

Fig. 1: Comparison of Random, Clustering, and Low-Rank Sensitivity sampling on the Default Credit Card dataset.

## 4.2 LLM Fine-tuning Experiments

### 4.2.1 Setting

**Models and datasets.** We fine-tune the standard instruction-tuned Llama3-8B model (Dubey et al., 2024) on three challenging downstream datasets: Grade-School Math (GSM8k) (Cobbe et al., 2021) with 7.47k training and 1.32k test samples, ViGGO (Juraska et al., 2019) with 5.1k training and 1.08k test samples, and SQL generation (Yu et al., 2018; Zhong et al., 2017) with 30k training and 1k test samples. This fine-tuning setup is widely used and has appeared in multiple research papers (Ashkboos et al., 2025; Chen et al., 2025; Nikdan et al., 2024). These tasks are specifically selected because the base models perform poorly on them, making them well-suited for fine-tuning. We closely follow the evaluation strategy of Ashkboos et al. (2025).

**Hyperparameters.** We largely adopt the training hyperparameters from the HALO code base (Ashkboos et al., 2025). For fine-tuning Llama3-8B-Instruct, we use the Adam optimizer for one epoch with learning rates $6 \times 10^{-6}$, $4 \times 10^{-5}$, and $3 \times 10^{-5}$ for GSM8k, ViGGO, and SQL, respectively. All dataset samples are encoded using the standard and efficient BERT embeddings (Devlin et al., 2019b). For clustering, we employ k-means++ and map each centroid to its closest sample in the dataset. The clustering procedure is repeated 10 times, and the best results are retained. For landmark selection in low-rank approximation, we employ leverage score sampling. By default, the number of clusters/landmarks is set to 20% of the total number of available samples, following the experimental setting of Axiotis et al. (2024). Regarding the parameters of Assumption 2.1, we tune the $\lambda$ value and pick the top performing one when applicable. Additionally, we set $\gamma = 0$, and compute $\alpha$ values in the embeddings space using Kernel Ridge Regression (KRR) with an RBF kernel to find the linear combination of landmark loss values.

**Baselines.** We consider three baselines: 1) *Full training*: where the data selection is skipped and the model is trained on the full dataset, 2) *Uniform sampling*, where the subset samples are selected uniformly at random, and 3) *Clustering-based sensitivity sampling* (Axiotis et al., 2024), which similar to our method, uses sensitivity sampling, but relies on clustering rather than low-rank approximation.

### 4.2.2 Results

Table 1: End-to-end fine-tuning validation accuracy on different baselines and datasets. BERT embeddings are used and $k$ is fixed to $25\%$ of the dataset. SS stands for Sensitivity Sampling.

| Dataset | GSM8k | | | ViGGO | | | SQL | | | Average | | |
|---|---|---|---|---|---|---|---|---|---|---|---|---|
| **Sampling Ratio** | **25%** | **12.5%** | **6.25%** | **25%** | **12.5%** | **6.25%** | **25%** | **12.5%** | **6.25%** | **25%** | **12.5%** | **6.25%** |
| *Uniform Sampling* | $67.7 \pm 0.3$ | $65.3 \pm 0.2$ | $63.5 \pm 0.5$ | $86.3 \pm 0.7$ | $68.3 \pm 4.1$ | $26.2 \pm 6.2$ | $75.6 \pm 0.5$ | $74.1 \pm 0.5$ | $66.2 \pm 3.5$ | 76.5 | 69.2 | 52.0 |
| *Clustering-based SS* | $\mathbf{70.2 \pm 0.1}$ | $66.6 \pm 1.2$ | $65.2 \pm 1.1$ | $86.6 \pm 2.8$ | $\mathbf{72.8 \pm 1.7}$ | $\mathbf{30.3 \pm 3.9}$ | $75.6 \pm 0.5$ | $73.7 \pm 0.5$ | $68.3 \pm 3.6$ | 77.5 | **71.0** | 54.6 |
| *Low-rank SS (ours)* | $68.4 \pm 0.1$ | $\mathbf{67.1 \pm 0.9}$ | $\mathbf{65.4 \pm 1.6}$ | $\mathbf{88.3 \pm 0.2}$ | $69.7 \pm 5.2$ | $28.8 \pm 1.1$ | $\mathbf{76.1 \pm 0.2}$ | $\mathbf{74.4 \pm 0.2}$ | $\mathbf{70.4 \pm 1.0}$ | **77.6** | 70.4 | **54.9** |
| *Full (100%)* | $69.3 \pm 0.5$ | | | $94.0 \pm 0.3$ | | | $79.9 \pm 0.5$ | | | 81.1 | | |

**Main results.** We begin by fine-tuning the Llama3-8B model on $25\%$, $12.5\%$, and $6.25\%$ of each dataset, selected using various sampling methods. Table 1 reports the validation accuracy of our method compared to the baselines. The results show that our method consistently outperforms uniform sampling. On average, it also achieves higher accuracy than clustering-based sensitivity sampling (Axiotis et al., 2024) in most cases, demonstrating the benefit of leveraging low-rank approximation for data selection.

**Runtime discussion.** The selection process for both cluster-based and low-rank sensitivity sampling requires forward passes on $k = 20\%$ of the dataset. Assuming a backward pass is twice as expensive as a forward pass (Kaplan et al., 2020), this corresponds to approximately $6.67\%$ of the total runtime for training on the full dataset.

**Study on dataset structure.** Here we analyze the training split of GSM8k to examine whether it exhibits a more clustered or low-rank structure. To this end, across a range of $k$ values, we perform the following experiments:

*i)* We cluster the per-sample embeddings into $k$ clusters and compute the average euclidean distance from each sample to its closest cluster center, and compare this with the average low-rank approximation error when representing the dataset using $k$ basis samples.

*ii)* We measure the average loss difference between each sample's true loss and that of its nearest cluster center, and compare it against the average difference between the true loss and the low-rank approximated loss.

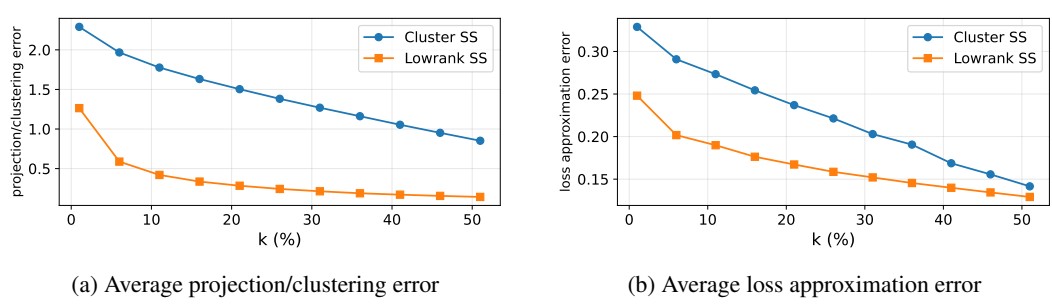

(a) Average projection/clustering error      (b) Average loss approximation error

Fig. 2: Comparison of cluster-based and low-rank sensitivity sampling methods on the GSM8k dataset using BERT embeddings. The values of $k$ are expressed as percentages of the entire dataset.

Figure 2 shows that, in both cases, the dataset yields a smaller error under the low-rank approximation, supporting our claim that the dataset (GSM8k in this case) is more aligned with a low-rank structure than with a purely clustered one.

**Average loss approximation quality.** We next investigate how well the (weighted) subset selected by each method approximates the average loss. We vary $k$ and $\lambda$, and in each case, select 2000 samples from the GSM8k dataset and measure the average loss approximation error ($\Delta(S)$, Section 3.1). Figure 3 presents heatmaps for both cluster-based and low-rank sensitivity sampling, showing that low-rank consistently achieves lower error. An interesting observation in the clustering case is that, at $\lambda = 1$, increasing the number of clusters degrades the approximation quality. This occurs because a large $\lambda$ causes the sampling score to be dominated by the geometric distance $r$, leading the algorithm to prioritize outliers over points from high-loss regions. When the number of clusters $k$ increases, the data space is partitioned more finely, reducing $r$ for inlier points and further biasing the selection toward outliers. Consequently, the selected subset becomes less representative of the overall distribution, resulting in poorer average loss approximation. A similar effect occurs for low-rank sampling at $\lambda \geq 100$, though these cases are omitted from the plots for clarity.

**Alternative objective and embedding.** Following Axiotis et al. (2024), we repeat our $12.5\%$ selection experiments on GSM8k, but replace the loss with the norm of per-sample gradients in Algorithm 1. Gradient norm serves as a proxy for capturing training dynamics (Axiotis et al., 2024). Figure 4 compares cluster-based and low-rank sensitivity sampling across different $\lambda$ values. The results indicate a slight advantage for low-rank sampling, which also appears more robust to the choice of $\lambda$, consistently outperforming uniform sampling for all values considered. Additionally, the same figure presents results for replacing BERT embeddings (Devlin et al., 2019b) with GTR-

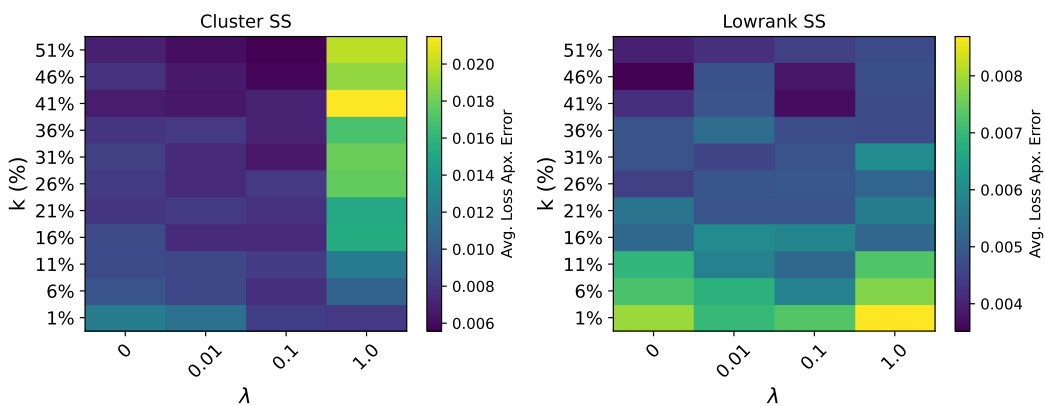

Fig. 3: Average loss approximation error across different $k$ and $\lambda$ values. In each case, 2000 (weighted) samples are selected from the GSM8k dataset, and the average of 100 trials is reported.

base embeddings (Ni et al., 2021). The results indicate that our positive findings remain consistent with these embeddings as well.

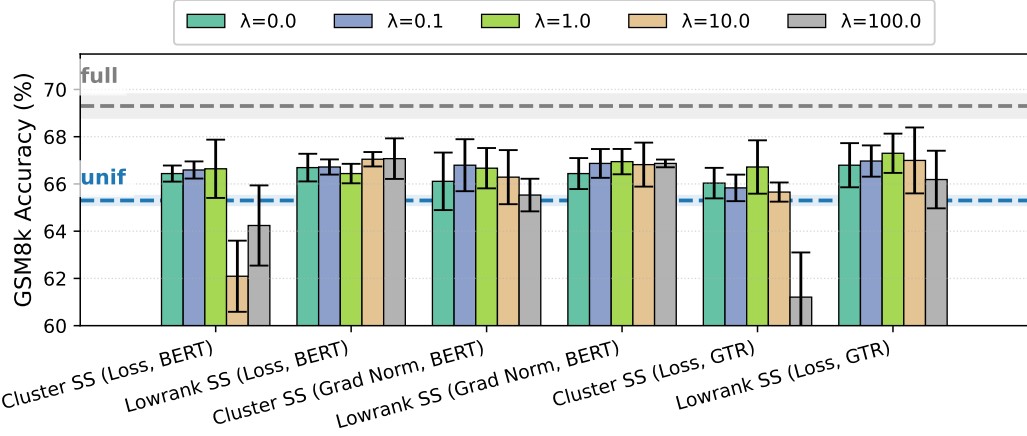

Fig. 4: Comparison of alternative objective functions (loss vs. gradient norm) and embedding functions, including BERT (Devlin et al., 2019b) and GTR (Ni et al., 2021), in terms of end-to-end validation accuracy, across various $\lambda$ choices. Experiments are conducted on the GSM8k dataset with $k$ fixed at 20%, and the selected subset size fixed at 12.5% of the dataset.

## 5 CONCLUSION

In this work, we introduced a novel data selection framework based on low-rank approximation, diverging from traditional clustering methods. We proposed a sensitivity sampling algorithm that constructs a small, weighted coreset to approximate the loss of the full dataset. Our main theoretical result, Theorem 2.2, provides a rigorous guarantee for this approach, with an error bound directly tied to the dataset's alignment with a low-rank structure.

Our empirical evaluations confirmed the practical benefits of this method. Across both a standard tabular dataset and challenging Llama3-8B fine-tuning on three tasks, our low-rank approach outperformed uniform sampling and clustering-based techniques in both approximation quality and downstream model performance. Our work provides a scalable, theoretically-grounded, and effective solution for data-efficient training, offering a robust alternative by leveraging the low-rank structure of data to identify the most informative samples.

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

## A  RELATED WORKS

**Deep learning and convolutional neural networks.** Deep learning methods, particularly convolutional neural networks (CNNs), have become the standard for large-scale image classification and related tasks. CNNs are especially powerful because they exploit spatial structure in images through convolutional filters, enabling models to learn hierarchical representations of features directly from raw pixels. While these models achieve state-of-the-art performance, they typically require very large labeled datasets to train effectively. This dependency on large-scale supervision motivates the study of techniques that can reduce the labeling burden without sacrificing performance, such as active learning and subset selection.

**Active learning.** A large body of work has examined the theoretical underpinnings of active learning. Classical results show that greedy selection is impossible in a fully agnostic setting (Dasgupta, 2004), yet refined analyses demonstrate stronger guarantees under assumptions such as realizability (Gonen et al., 2013) or bounded disagreement coefficients Hanneke (2007). Other approaches justify greedy strategies in the batch setting via importance sampling Ganti & Gray (2012). Although these works provide rigorous guarantees, they do not address the large-scale deep learning problems that motivate our study.

Complementing these theoretical contributions, several algorithms have been designed specifically for CNNs. Wang et al. (2017) propose auto-labeling of confident predictions while querying uncertain points, and Stark et al. (2015) develop a method tailored for CAPTCHA recognition. These

CNN-oriented techniques succeed in narrow domains but do not scale to general-purpose image classification tasks.

A different stream of research views active learning through the lens of optimization. Formulations that balance uncertainty and diversity often cast the problem as a discrete program with convex relaxations (Elhamifar et al., 2013; Yang et al., 2015; Guo, 2010), but these require $n^2$ variables for $n$ data points, making them impractical for large datasets. More specialized efforts adapt active learning to k-nearest neighbors, naive Bayes, or logistic regression (Wei et al., 2015; Hoi et al., 2006; Guo & Schuurmans, 2007; Yu et al., 2006). Within this optimization-oriented family, Demir et al. (2011) propose a two-stage approach that first filters uncertain points and then enforces diversity. Our method is closely related but is the first to be applied directly to CNNs. Indeed, the most similar efforts are by Joshi et al. (2010) and Wang & Ye (2015): the former introduces a related optimization problem without theory, while the latter minimizes maximum mean discrepancy. Neither is designed or tested for CNNs, whereas our framework builds on these ideas by introducing the notion of coreset loss, providing both a theoretical foundation and practical applicability to deep models.

Finally, classical acquisition strategies remain an influential part of the literature. Early surveys such as Settles (2009) summarize information-theoretic approaches (MacKay, 1992), ensemble-based methods (McCallum & Nigam, 1998; Freund et al., 1997), and uncertainty-driven heuristics (Tong & Koller, 2001; Joshi et al., 2009; Li & Guo, 2013). In particular, uncertainty-based sampling focuses on querying ambiguous points, using entropy (Joshi et al., 2009) or margin-based distances (Tong & Koller, 2001; Brinker, 2003). Bayesian active learning has also been widely studied, traditionally with Gaussian processes to estimate error reduction or predictive improvement (Roy & McCallum, 2001; Kapoor et al., 2007). While powerful in small-scale settings, these approaches do not scale to modern CNNs. Recent work reinterprets dropout as approximate Bayesian inference (Gal & Ghahramani, 2016), extending Bayesian methods to deep architectures, with follow-up experiments on modest datasets (Gal et al., 2017). Our experiments, however, show that these methods remain limited in batch settings and fail to scale effectively.

# B  MISSING PROOFS

**Theorem 2.2.** *[Coreset Guarantee for Loss Approximation] Let $D$ be a dataset of $n$ points with an embedding $E$, and suppose the loss function $\ell$ satisfies Assumption 2.1 with constants $\gamma, \lambda$. Let*

$$\Phi_k(D) = \min_{\substack{D_k \in \mathbb{R}^{n \times m} \\ \mathrm{rank}(D_k) \leq k}} \|D - D_k\|_F^2$$

*denote the best rank-$k$ approximation cost of $D$. Then there exists a randomized algorithm that constructs a weighted subset $S \subseteq D$ of size $s = \mathcal{O}\left(\frac{1}{\varepsilon^2}\right)$ with weights $w(x)$ such that, with probability at least $0.9$,*

$$\left| \sum_{x \in D} \ell(x) - \sum_{x \in S} w(x)\, \ell(x) \right| \leq \varepsilon \left( \sum_{x \in D} \ell(x) + \gamma \|D\|_F^2 + \gamma k |D| \max \ell + 2\lambda\, \Phi_k(D) \right). \quad (1)$$

*Equivalently, the weighted average loss on $S$ is within a $(1 \pm \varepsilon)$ factor of the true average loss, up to an additive term proportional to $\Phi_k(D)/n$.*

*Proof.* Let

$$L := \sum_{x \in D} \ell(x)$$

be the total loss over the dataset $D$, and define

$$\Phi_k(D) = \min_{\mathrm{rank}(V) \leq k} \|D - V\|_F^2,$$

the best rank-$k$ approximation error of $D$. For every point $x \in D$, let $v(x) = \mathrm{Proj}(x, V)$ be the projection of $x$ onto the chosen low-rank approximation $V$ and let $r(x) = x - \mathrm{Proj}(x, V)$ be the orthogonal complement so that $x = v(x) + r(x)$. By the Lipschitz condition (with constant $\lambda$), we have for every $x \in D$:

$$\left| \ell(x) - \ell(v(x)) \right| \leq \lambda \cdot \|r(x)\|_2^2.$$

Suppose we have $v(x) = \alpha_1 v_1 + \ldots + \alpha_k v_k$. Then we also have

$$\left|\ell(v(x)) - (\alpha_1^2 \ell(v_1) + \ldots + \alpha_k^2 \ell(v_k))\right| \leq \gamma \left(|\alpha_1^2 - 1|\ell(v_1) + \ldots + |\alpha_k^2 - 1|\ell(v_k)\right),$$

where $v(x) = \alpha_1 v_1 + \ldots + \alpha_k v_k$. Hence by triangle inequality, we have

$$\ell(x) \leq (\alpha_1^2 \ell(v_1) + \ldots + \alpha_k^2 \ell(v_k)) + \gamma \left(|\alpha_1^2 - 1|\ell(v_1) + \ldots + |\alpha_k^2 - 1|\ell(v_k)\right) + \lambda \|r(x)\|_2^2$$
$$\leq (\gamma + 1)(\alpha_1^2 \ell(v_1) + \ldots + \alpha_k^2 \ell(v_k)) + \gamma k \cdot \max_k \ell(v_k) + \lambda \|r(x)\|_2^2$$

and

$$(\alpha_1^2 \ell(v_1) + \ldots + \alpha_k^2 \ell(v_k)) \leq \ell(x) + \gamma \left(|\alpha_1^2 - 1|\ell(v_1) + \ldots + |\alpha_k^2 - 1|\ell(v_k)\right) + \lambda \|r(x)\|_2^2$$
$$\leq \ell(x) + \gamma(\|x\|_2^2 + k) \cdot \max_k \ell(v_k) + \lambda \|r(x)\|_2^2.$$

Let $\xi \geq \max_k \ell(v_k)$. Then we next define the sensitivity score for each $x \in D$ as the following:

$$\sigma(x) := (\gamma + 1)(\alpha_1^2 \ell(v_1) + \ldots + \alpha_k^2 \ell(v_k)) + \gamma k \xi + \lambda \|r(x)\|_2^2,$$

where $v(x) = \alpha_1 v_1 + \ldots + \alpha_k v_k$. Assign the sampling probability by normalizing these scores:

$$p(x) := \frac{\sigma(x)}{T}, \quad \text{where} \quad T := \sum_{y \in D} \sigma(y).$$

We now select $s$ independent samples (with replacement) from $D$ according to $p(x)$; define $S = \{x_1, \ldots, x_s\}$ as the resulting multiset. For every sample $x \in S$, define its weight as

$$w(x) := \frac{1}{s\, p(x)}.$$

Hence, the weighted loss estimator is

$$Z := \sum_{x \in S} w(x)\, \ell(x).$$

Through the linearity of expectation,

$$\mathbb{E}\big[\ell(x)w(x)\big] = \sum_{x \in D} p(x) \cdot \frac{\ell(x)}{s\, p(x)} = \frac{1}{s} \sum_{x \in D} \ell(x) = \frac{L}{s},$$

so $\mathbb{E}[Z] = L$; that is, the estimator is unbiased.

For a single sample, let

$$X = \ell(x)w(x) = \frac{\ell(x)}{s\, p(x)}.$$

Then its second moment is

$$\mathbb{E}[X^2] = \sum_{x \in D} p(x) \left(\frac{\ell(x)}{s\, p(x)}\right)^2 = \frac{1}{s^2} \sum_{x \in D} \frac{\ell(x)^2}{p(x)}.$$

Substituting $p(x) = \sigma(x)/T$, we get the following

$$\mathbb{E}[X^2] = \frac{T}{s^2} \sum_{x \in D} \frac{\ell(x)^2}{(\gamma + 1)(\alpha_1^2 \ell(v_1) + \ldots + \alpha_k^2 \ell(v_k)) + \gamma k \xi + \lambda \|r(x)\|_2^2}.$$

Since $\ell(x) \leq (\gamma + 1)(\alpha_1^2 \ell(v_1) + \ldots + \alpha_k^2 \ell(v_k)) + \gamma k \xi) + \lambda \|r(x)\|_2^2$, it follows that

$$\frac{\ell(x)^2}{(\gamma + 1)(\alpha_1^2 \ell(v_1) + \ldots + \alpha_k^2 \ell(v_k)) + \gamma k \xi + \lambda \|r(x)\|_2^2} \leq \ell(x).$$

Thus,

$$\mathbb{E}[X^2] \leq \frac{T}{s^2} \sum_{x \in D} \ell(x) = \frac{L\,T}{s^2}.$$

Summing over all $s$ samples, we have

$$\sum_{i=1}^{s} \mathbb{E}[X_i^2] \leq \frac{L\,T}{s}.$$

Using the bound $T \leq L + \gamma(\|D\|_F^2 + k|D|)\xi + \lambda R$, where

$$R := \sum_{x \in D} \|r(x)\|_2^2,$$

we obtain

$$\sum_{i=1}^{s} \mathbb{E}[X_i^2] \leq \frac{\left(L + \gamma(\|D\|_F^2 + k|D|)\xi + \lambda R\right)^2}{s}.$$

For any point $x \in D$, its weighted contribution is

$$\ell(x)w(x) = \frac{\ell(x)}{s}\frac{1}{p(x)} = \frac{\ell(x)}{s}\frac{T}{\sigma(x)}.$$

Since $\ell(x) \leq \sigma(x)$, it follows that

$$\ell(x)w(x) \leq \frac{T}{s} \leq \frac{L + \gamma(\|D\|_F^2 + k|D|)\xi + \lambda R}{s}.$$

Thus, if we set

$$M := \frac{L + \gamma(\|D\|_F^2 + k|D|)\xi + \lambda R}{s},$$

then $|X_i| \leq M$ for every sample.

Let

$$Z = \sum_{i=1}^{s} X_i = \sum_{x \in S} w(x)\,\ell(x).$$

By Bernstein's inequality, for any $t > 0$,

$$\Pr\Big(|Z - L| \geq t\Big) \leq \exp\left(-\frac{t^2}{2\sum_{i=1}^{s}\mathbb{E}[X_i^2] + \frac{2}{3}M\,t}\right).$$

Set $K = \|D\|_F^2 + k|D|$ so that $\gamma(\|D\|_F^2 + k|D|)\xi = \gamma K\xi$ and set

$$t := \varepsilon\Big(L + \gamma K\xi + \lambda\,\Phi_k(D)\Big).$$

Then,

$$\Pr\left(|Z - L| \geq \varepsilon\Big(L + \gamma K\xi + \lambda\,\Phi_k(D)\Big)\right) \leq \exp\left(-\frac{\varepsilon^2\big(L + \gamma K\xi + \lambda\,\Phi_k(D)\big)^2}{2\frac{(L + \gamma K\xi + \lambda R)^2}{s} + \frac{2}{3}\frac{L + \gamma K\xi + \lambda R}{s}\,\varepsilon\big(L + \gamma K\xi + \lambda\,\Phi_k(D)\big)}\right).$$

By choosing

$$s = \left\lceil \varepsilon^{-2}\Big(2 + \frac{2\varepsilon}{3}\Big)\right\rceil,$$

the exponent can be made sufficiently large so that the probability of failure is below $0.1$. That is, with probability at least $0.9$,

$$\left|\sum_{x \in D} \ell(x) - \sum_{x \in S} w(x)\,\ell(x)\right| \leq \varepsilon\Big(L + \gamma K\xi + \lambda\,\Phi_k(D)\Big).$$

$\square$