# OpenReview forum: "Data Selection and Active Learning via Low-Rank Approximation"
_ICLR.cc/2026/Conference — ICLR 2026 Conference Withdrawn Submission_

### Official Review · Reviewer_dhyi · 2025-10-19

**Soundness:** 2
**Presentation:** 2
**Contribution:** 2
**Rating:** 2
**Confidence:** 3

**Summary:**

This paper presents a data selection method based on low-rank approximation. It derives an error bound for loss approximation that depends on how well the dataset aligns with a low-rank structure. Based on this bound, a sensitivity sampling method is developed. The effectiveness of this method is validated through quantitative experiments and analysis on credit card and LLM fine-tuning datasets.

**Strengths:**

1. Theoretical foundation. The method is grounded in a theorem that provides formal performance guarantees under certain assumptions.

2. Detailed comparative analysis. The paper includes a detailed comparison with a clustering-based baseline and provides analytical discussions on the results.

**Weaknesses:**

1. Unjustified assumptions. While the Lipschitz-like assumption for deviations along the orthogonal directions is reasonable, the second part of Assumption 2.1 (the basis decomposition assumption) lacks justification. If it holds, the loss function must approximately satisfy $\ell(\sum_{i} \alpha_i v_i) \approx \sum_{i} \alpha_i^2 \ell(v_i)$, implying strong constraints such as $\ell(v_1+v_2) \approx \ell(v_1)+\ell(v_2)$ and $\ell(\alpha_1v_1) = \alpha_1^2 \ell(v_1)$. Further explanation or empirical evidence is needed to justify this assumption.

2. Unclear algorithm and limited guidance for implementation. Sec. 3.3 does not provide sufficient detail for reproducibility. Although Algorithm 1 helps, there is limited discussion on how to set hyperparameters (target rank $k$, error parameter $\epsilon$, constants $\lambda$ and $\gamma$) and how these choices influence the data selection results.

3. Limited evaluation and modest performance. The evaluation includes only two baseline methods, despite extensive prior work on data selection and active learning. Moreover, the proposed method performs comparably or worse than one of the baselines in LLM fine-tuning experiments, raising concerns about its effectiveness and generalizability.

**Questions:**

1. In the introduction, why "processing data in batches" would induce "correlations among selected samples", and why is this considered harmful?
2. In Fig. 1(b), why does test accuracy not increase as the sample size grows for the clustering-based method?
3. In Sec. 4.2.1, "landmark selection" and "leverage score sampling" seem related to the proposed method but were not introduced earlier.
4. In Sec. 4.2.2, it would be beneficial to include the overhead of both the proposed and baseline methods in the runtime discussion.
5. In Sec. 4.2.2, clustering with $k$ centers and low-rank approximation with $k$ bases are not directly comparable. For a dataset $D\in \mathbb{R}^{n\times m}$ where $n >> m$, only $m$ bases are required for zero approximation error, while clustering would require many more centers.
6. In Fig. 3, the color scales of the two heatmaps should be unified to facilitate comparison.

---

### Official Review · Reviewer_68DS · 2025-10-30

**Soundness:** 2
**Presentation:** 2
**Contribution:** 2
**Rating:** 4
**Confidence:** 3

**Summary:**

This paper introduces a novel data selection technique leveraging low-rank approximation and residual sampling. The authors propose a sensitivity sampling algorithm designed to construct a small, weighted coreset that effectively approximates the loss of the full dataset. The efficacy of this method is evaluated on the "Default of Credit Card Clients" dataset and a Large Language Model (LLM) fine-tuning task.

**Strengths:**

The core idea of utilizing low-rank approximation and residual sampling contributes to construction of coreset. This algebraic approach presents an interesting alternative to more common geometric methods like clustering.

The application of this technique to LLM fine-tuning is particularly compelling and shows potential for extension to other tasks within the LLM domain.

**Weaknesses:**

1.  **Clarification of Novelty**: While applying low-rank approximation to data selection is interesting, the core techniques have been explored in related contexts. Low-rank methods are well-established for feature dimensionality reduction, and coreset construction using sensitivity sampling has been previously combined with geometric approaches like k-means clustering.

2. **Paper Organization**: The structure of the paper could be improved for better readability. The current layout, which presents assumptions and theoretical results before restating the problem setting in Section 3.2, leads to some repetition and could be confusing for the reader.

3. **Clarity of Theoretical Framework**: Several notations and assumptions within the theoretical sections are not fully clarified, which may hinder a complete understanding of the theoretical contributions (see Questions below).

**Questions:**

**Main Concerns**

1. **Basis Decomposition** (Line 121): The decomposition of $y$ into a linear combination of basis vectors assumes a specific form for the coefficients $\alpha_i$. However, this formulation requires the basis vectors $v_i$ to be orthogonal which is not imposed.

2.	**Rationale for Assumption 2.1**: Could you provide further justification for the second inequality in Assumption 2.1? Specifically, in the case where all $\alpha_i = 1$ , the assumption implies the loss can be decomposed into a sum of losses for each basis vector. Does this require loss function to be specific form?


3.	**Clarification of Theorem 2.2**: The statement of Theorem 2.2 could be made clearer.

     a)	Can the success probability of "at least 0.9" be generalized to the standard "$1-\delta$"?

     b)	Could you please provide explicit definitions for the notations |D| (is it the number of samples, n?) and max $\ell$?


4.	**Notation in Algorithm 1**: The symbol $\xi$ is used in Algorithm 1 without a prior definition. Please clarify its meaning.

**Minor Concerns**

5.	**Citation Style (Abstract)**: The citation "Sener and Savarese [ICLR 2018]" does not follow standard academic formatting. Please consider using a consistent, standard citation style (e.g., author-year or numerical).

6.	**Redundant Statements (Lines 83 & 137)**: The statements on these lines appear to be repetitive. Consolidating them would improve the manuscript's conciseness.
7.	**Notation Order (Line 143)**: The order of $\lambda$ and $ \gamma$ appears to be inconsistent with previous usage. Please check for consistency.
8.	**Inconsistent Dimensionality Notation:** In Theorem 2.2, the data matrix $D$ is described as $n \times m$, where m is the data point dimension. However, the assumptions refer to the dimension as d. Please use consistent notation for the data dimensionality throughout the paper.
9.	**Inconsistent Input Notation:** The input variable is denoted as $y $ in the assumptions but as $x$ in the theorem. Please unify this notation.
10.	**Word Choice (Line 200)**: The phrase "efficient and effective" is a common cliché. Using a more specific or concise alternative could strengthen the statement.

---

### Official Review · Reviewer_kwSv · 2025-10-30

**Soundness:** 2
**Presentation:** 2
**Contribution:** 3
**Rating:** 4
**Confidence:** 4

**Summary:**

This paper presents a low-rank sensitivity sampling framework, which first builds a rank-k sketch of data embeddings and scores points via three components before sampling a weighted subset. The main theoretical contribution establishe that a subset of size $O(\varepsilon^2)$ suffices to achieve a $1\pm\varepsilon$ relative loss approximation, up to an additive term proportional to $\Phi_k(D)/n$. Experiments on tabular datasets and LLM fine-tuning across various tasks validate the efficacy of the proposed sampling strategy.

**Strengths:**

+ The motivation of preserving global spectral structure is well established.
+ Theorem 2.2 provides an explicit sample-size vs. error trade-off
+ Combine projection loss, base loss, and residual loss when scoring data points is interesting and effective

**Weaknesses:**

- Lack in-depth analysis into cases where low-rank sensitivity sampling underperforms clustering sensitivity sampling in Table 1.
- The baseline only includes uniform sampling and clustering-based sensitivity sampling; many classic coreset selection methods are missing, such as k-center, GraphCut, and facility-location. The authors are suggested to systematically review state-of-the-art coreset selection methods (especially those published in ICLR 2024) and compare the proposed method with them.
- The runtime analysis lacks the discussion about the additional cost of embedding extraction and performing SVD. Hyperparameter search time is also unaccounted.
- Spectral geometry may differ across modalities and model architectures. Current evidence centers on one LLM family and a small set of text embeddings. More modalities and model architectures should be tested to validate the robustness of the method.

**Questions:**

- When will the low-rank method beat the clustering method, and how to select them before model training?
- Will the low-rank method work well on different modalities (e.g. vision/audio) and different architectures (e.g. T5)?
- Compare the method with the representative coreset selection method and analyze the results,
- Provide a detailed analysis of the runtime cost.

---

### Official Review · Reviewer_nPAR · 2025-11-07

**Soundness:** 2
**Presentation:** 2
**Contribution:** 2
**Rating:** 0
**Confidence:** 5

**Summary:**

This paper introduces a new method for dataset subset selection based on a low-rank approximation (computed via SVD) of the dataset. Under certain assumptions on the loss function, the paper suggests that the selected subset achieves similar loss as the full dataset. Empirical comparisons on a tabular dataset and a LLM fine-tuning example suggest that the proposed method outperforms random selection and a clustering-based baseline.

**Strengths:**

**S1) Exploiting low-rank structure as a means of data selection is an interesting and promising idea**

The experimental results show improvements in the coreset selection problem over baselines.

**Weaknesses:**

**W1) Algorithm implementation needs more clarity**

My main concern with the algorithm is that it requires computing $\ell(v_j)$ for $j = 1, \dotsc, k$. I understand that the subspace basis $v_1, \dotsc, v_k$ comes from running truncated SVD on the data matrix $X \in \mathbb{R}^{n \times d}$. In the case that $\ell$ does not depend on labels (e.g., $\ell(x) = \|x\|_2^2$), I can see how you compute $\ell(v_j)$. For losses that depend on a data label $y$, though, how do you compute $\ell(v_j)$? Most likely $v_j$ is not an entry in the dataset, so it does not have an associated label. Please clarify this critical part of the algorithm.

**W2) Paper writing is unclear and confusing in many places**

The paper writing is confusing and unclear in several key places, making it challenging to evaluate the contributions of the paper. See “Questions” section below. Some other places that need clarification are:

- Line 296: what does “heterogeneous” refer to hear?

- Line 296-297: if the Default of Credit Card Clients dataset is a “standard benchmark,” please provide citations to justify this claim


**W3) Assumptions need more justification**

Please clarify how Assumption 2.1 is or isn’t satisfied by common loss functions. In Line 133, the authors write that “these assumptions are natural in many machine learning settings,” without providing any formal mathematical justification. Lines 133-141 seem very hand-wavy for such a key claim.

**W4) Active learning claim may be overstated**

The title of the paper includes the phrase “active learning,” yet the paper does not provide any theoretical or experimental support for any active learning setting. While active learning and data selection are often related, the paper does not describe how the proposed algorithm can be used in an active learning context.

**W5) Missing important baselines**

While the paper claims that low-rank approximation may be more useful than clustering-based approaches, the paper fails to compare against the seminal core-set selection approach by Sener and Savarese (2018) or other related and widely-cited works such as Selection-via-proxy (Coleman et al., 2020).

**W6) Concerns on “Study on dataset structure”**

I'm not sure that the “Study on dataset structure” is a meaningful comparison. For clustering, the paper reduces the entire dataset down to a ($k \times d$) matrix, requiring $k \times d$ parameters. In contrast, for low-rank approximation, the paper reduces the dataset down to a ($n \times k$) matrix of weights and a ($k \times d$) matrix of basis vectors, for a total of $k \times (n+d)$ parameters. To be more fair, I think the comparison should use the same number of parameters for each method:

- clustering: $k_\text{cluster} \times d$

- low-rank approximation: $k_\text{low-rank} \times (n + d)$


**W7) Irrelevant related works section**

The related works section in the appendix makes claims that are simply not substantiated:

- “Our method is closely related but is the first to be applied directly to CNNs.” - the paper does not run any experiments on CNN models.

- “Our experiments, however, show that these methods remain limited in batch settings and fail to scale effectively.” - the paper does not run any experiments for the methods described earlier in the paragraph.


I am concerned that these claims are instead paraphrased directly from the Sener and Savarese (2018) paper. Please see my discussion in the “Details Of Ethics Concerns.”

**W8) Other minor concerns**

- Please add error bars to Figure 1

- Please be consistent with notation. If $D = \{(x_i, y_i)\}_{i=1}^n$, then you should always write $(x,y) \in D$ instead of just $x \in D$ (e.g., line 275).

**Questions:**

**Q1)** For most of the paper, $k$ refers to the rank of the subspace. However, in Section 4.2.2, the authors write “$k$ is fixed to 25% of the dataset” (line 371) and “$k$ = 20% of the dataset” (line 385). First, these two statements are contradictory. Second, how is “X% of the dataset” related to the rank of the subspace?

**Q2)** In Theorem 2.2, the value 0.9 (for the probability) seems rather arbitrary. Why 0.9? Does this hold for higher probability values?

**Q3)** In Theorem 2.2, the last sentence reads, “Equivalently, the weighted average loss on $S$ is within a $(1+\epsilon)$ factor of the true average loss, up to an additive term proportional to $\Phi_k(D)/n$. What about the $\|D\|_F^2$ term? Isn’t that term also additive?

**Q4)** In the experiments on the Credit Card dataset, why do you need to first fit a model over the full training set “to obtain per-point logistic losses”?

**Q5)** In Section 4.2.1 (hyperparameters), why do you set $\gamma = 0$, and why/how do you use Kernel Ridge Regression to find loss values? Please clarify and explain.

**Q6)** In the proof of Theorem 2.2, it seems to be assumed that the loss $\ell$ is always nonnegative. If so, please state this assumption clearly. Is there any way to generalize the result to handle losses which can be negative?

**Q7)** In the proof of Theorem 2.2, could you please clarify how you go from line 766 to line 767? It’s not clear to me where the term $\|x\|_2^2$ comes from.

**Details Of Ethics Concerns:**

I am concerned that this paper paraphrases significantly from two related works. While the words are not copied directly, the structure and claims are exceptionally similar. Furthermore, some of the claims do not appear to even apply to this paper.

1. Appendix A (Related Works) seems to be largely paraphrased from the related works section of Sener and Savarese (2018). As mentioned in (W7) above, this manuscript makes two claims that appear in the Sener and Savarese (2018):

    - “Our method is closely related but is the first to be applied directly to CNNs.”

    - “Our experiments, however, show that these methods remain limited in batch settings and fail to scale effectively.”


    However, this paper does discuss CNNs in any meaningful way, nor does it compare to the methods described in the related works section.

2. The introduction is extremely similar to the Axiotis et al. (2024) paper, with the exception of the final paragraph in the introduction. The paragraph ordering, word choice, phrasing, and citations are all very similar.


While the wording is not 100% identical, I believe that this style of paraphrasing is at least borderline plagiarism. As defined by the [Merriam-Webster dictionary](https://www.merriam-webster.com/dictionary/plagiarizing), plagiarism is

> to steal and pass off (the ideas or words of another) as one's own : use (another's production) without crediting the source.

I would argue that this paper passes off the ideas (and to a large extent, the words as well) from the Sener and Savarese (2018) and the Axiotis et al. (2024) papers. Furthermore, this paper certainly does not properly credit the two aforementioned citations for the ideas in the intro and related works.

---

> ### Author Response · Authors · 2025-12-03
>
> > I am concerned that this paper paraphrases significantly from two related works. While the words are not copied directly, the structure and claims are exceptionally similar. Furthermore, some of the claims do not appear to even apply to this paper. Appendix A (Related Works) seems to be largely paraphrased from the related works section of Sener and Savarese (2018).
>
> > The introduction is extremely similar to the Axiotis et al. (2024) paper, with the exception of the final paragraph in the introduction. The paragraph ordering, word choice, phrasing, and citations are all very similar.
>
> > While the wording is not 100% identical, I believe that this style of paraphrasing is at least borderline plagiarism.
>
> > I would argue that this paper passes off the ideas (and to a large extent, the words as well) from the Sener and Savarese (2018) and the Axiotis et al. (2024) papers. Furthermore, this paper certainly does not properly credit the two aforementioned citations for the ideas in the intro and related works.
>
> We appreciate the reviewer’s concern regarding the similarity in framing to prior works. Our paper studies active learning via data selection, which is the same problem setup and motivation as Sener & Savarese (2018) and Axiotis et al. (2024), so it is natural that the high-level context and problem motivation resemble these works. We also cite both papers prominently—beginning in the abstract and continuing throughout the introduction and related-work sections—to make clear how our work builds on and differs from prior research.
>
> That said, we recognize that certain parts of the exposition could be rephrased to better highlight our perspective and avoid any unintended overlap in presentation. We will take the reviewer’s feedback as an opportunity to refine the introduction and related-work sections in the next version, so that the distinctions are even clearer and the narrative is more aligned with our contributions.

---

### Note · Authors · 2025-12-03

I have read and agree with the venue's withdrawal policy on behalf of myself and my co-authors.